# Structures and Biological Activities of Alkaloids Produced by Mushrooms, a Fungal Subgroup

**DOI:** 10.3390/biom12081025

**Published:** 2022-07-24

**Authors:** Jesús G. Zorrilla, Antonio Evidente

**Affiliations:** 1Allelopathy Group, Department of Organic Chemistry, Institute of Biomolecules (INBIO), School of Science, University of Cadiz, C/Republica Saharaui, s/n, 11510 Puerto Real, Spain; 2Department of Chemical Sciences, University of Naples “Federico II”, Complesso Universitario Monte Sant’Angelo, Via Cintia 4, 80126 Napoli, Italy; evidente@unina.it

**Keywords:** fungi, mushrooms, alkaloids, structure, biological activity, structure-activity relationship, potential practical application

## Abstract

Alkaloids are a wide family of basic *N*-containing natural products, whose research has revealed bioactive compounds of pharmacological interest. Studies on these compounds have focused more attention on those produced by plants, although other types of organisms have also been proven to synthesize bioactive alkaloids, such as animals, marine organisms, bacteria, and fungi. This review covers the findings of the last 20 years (2002–2022) related to the isolation, structures, and biological activities of the alkaloids produced by mushrooms, a fungal subgroup, and their potential to develop drugs and agrochemicals. In some cases, the synthesis of the reviewed compounds and structure−activity relationship studies have been described.

## 1. Introduction

Natural sources have a great diversity of *N*-containing compounds. Numerous studies have been performed on the isolation and chemical and biological characterization, and these studies are still increasing. These investigations have also confirmed that families such as alkaloids [1], peptides [2], phenoxazines [3], amines [4], or nitrogenous sesquiterpenoids [5] could show outstanding activities of pharmacological or agronomic interest. The alkaloid family is one of the most relevant of these, given its production by a wide range of living beings, its structural variety, as well as the biological activities that have been discovered a long time ago. 

Alkaloids are a large group consisting of diverse subgroups of natural products that are most extensively studied in plants. Some examples of well-known alkaloids of a vegetal origin are morphine, which possesses common anesthetic and pain reliever activities [6]; caffeine, which is a stimulant in commonly consumed beverages [7]; or nicotine, which is an addictive constituent in tobacco [8]. Among the plant alkaloids, there is a large group produced by hundreds of species of Amaryllidaceae, for which their chemistry and biological activities have also been reported on in previous reviews [9]. These alkaloids have an assumed importance, not only for their chemistry, but also for their several biological activities [10,11,12,13,14,15,16]. Lycorin is the main Amaryllidaceae alkaloid, which has been known for a long time as lycorine, and has essentially been studied for its anticancer activity as well as for its natural and synthetic analogs and close isocarbostyryls [17,18,19,20]. However, several studies have shown the presence of alkaloids with promising medicinal properties in other types of organisms, including animals, insects (an animal subgroup), marine sources, bacteria, fungi, and mushrooms (one of the subgroups of fungi). Figure 1 provides an overview of the type of organisms involved in studies on alkaloids. From these data, it is possible to conclude that mushrooms are one of the least studied sources for alkaloids, which are only surpassed by lichens. Nevertheless, a sufficient number of scientific articles have reported the isolation and the biological activities of diverse mushroom alkaloids. The example of psilocybin and its metabolic product psilocin could be hilighted, which are two of the most studied hallucinogenic compounds from the psilocybin mushrooms [21]. 

Thus, on the basis of these results, we report the biological and chemical characterization of mushroom alkaloids, as this source has been lesser studied than the others.

This review is focused on studies carried out on mushrooms over the last 20 years in relation to the alkaloids they produce. Considering Figure 1, 390 of the 506 references available in the literature for this topic, that is, 77%, were published throughout this period. This study intends to highlight the most significant developments found in the reviewed period, thereby giving rise to take perspective to carry out new research on this promising field. In some cases, the synthesis reported in the literature for some of the reviewed compounds will be highlighted. This is the case for laccarin, an alkaloid isolated at a low yield from the mushroom *Laccaria vinaceoavellanea*, which can become available through the enantioselective synthesis developed by Bower et al. (2007) [22]. 

The bibliography was selected from the database SciFinder by combining the keywords “alkaloid” and “mushroom”. The search was restricted to the period of 2002–2022. Additionally, some references were collected through complementary searches through SciFinder or Google Scholar. After a critical reading of the articles, 144 articles were selected and their main results and conclusions are included in this review.

The review is divided into subheadings considering the carbon skeleton of the reviewed compounds, in their chronological order of publication. Moreover, this review covers diverse structure-activity relationship (SAR) studies carried out during the reviewed period. These studies are generally based on the synthesis and evaluation of the bioactivity of a number of structural analogs, providing the best cases for the specific structural modifications that improve the activity levels. The study by Yuan et al. (2017) [23] represents a recent example of this kind of this study, and also provides the enantioselective synthesis of the already-known mushroom alkaloid lysergol.

## 2. New Alkaloids Found in Mushroom since 2002

Section 2 reports, in detail, the new alkaloids discovered in mushrooms during 2002–2022 (30 May). Given the larger number of compounds found for β-carbolines, pyrroloquinolines, pyrroles, and indoles, they have been grouped and described independently according to their carbon skeleton in Section 2.1, Section 2.2, Section 2.3 and Section 2.4. The alkaloids that have not been grouped are described in Section 2.5 in chronological order according to their year of discovery.

A structural consideration to take into account is that alkaloids are natural products whose nitrogen atom has basic properties. By extension, compounds that differ in this respect but are biogenetically related to them could be included [24], and this classification has been adopted in this review.

Thus, Table 1 shows the new alkaloids and related compounds produced by mushrooms discovered during the period covered by the review, together with their isolation source and the biological activities that were described for them.

### 2.1. β-Carboline Alkaloids

β-Carboline alkaloids are known for their various biological activities, including their antioxidant, antimicrobial, antiparasitic, antiviral, antitumor, hallucinogenic, and DNA intercalation activity, among others [30]. Norharman, and its methylated derivative harman (**1** and **2**, Figure 2), are among the most studied alkaloids from this family. They are normal endogenous body constituents that possess pharmacological properties, including cytotoxicity [76,77]. However, these two compounds might cause Parkinson’s and cancer [78]. These alkaloids have also been found in tobacco smoke and in other diverse plant species, as well as in food and drink [78,79,80]. Moreover, they are also produced by bacteria [77,81] and fungi [82]. Their occurrence in mushrooms has also been reported, and has been found in 27 species of the genus *Hygrophorus* [27] and in the *Psilocybe* species [83]. Harmine and harmaline (**3** and **4**, Figure 2) represent other known mushroom alkaloids with pharmacological properties [84,85]. 

Canthin-6-one alkaloids **5**–**8** (Figure 2) were the first discovered β-carboline alkaloids in the review period and were isolated from the fruiting bodies of *Boletus curtisii* [25]. Canthin-6-one (or canthinone, **9**, Figure 2) alkaloids are a subclass of β-carboline alkaloids that contain an additional D-ring [86]. Alkaloids **5**–**8** are characterized by the presence of a sulfur atom in their structure. In particular, they are close to canthin-6-one (**9**), but differ from it because of the presence of a thiomethyl group in different positions [25]. The same authors also reported the first isolation of canthin-6-one (**9**) outside of higher plants. Compound **9** has anti-fungal, anti-parasite, and cytotoxic properties [86,87]. As no activities were reported for thiomethylated alkaloids **5**–**8**, extensive studies on their pharmacological activities would be of interest.

The harmane derivatives β-carboline-1-propanoic acid and 2-methyl-β-carbolinium-1-propanoate (**10** and **11**, Figure 2), the latter as a new compound, were also isolated from *B. curtisii* [25]. Compound **10** was also found in *Cortinarius infractus* [88] and in the plant kingdom [89,90,91], including its tentative identification in extracts from the matrix plants of the Ayahuasca tea beverage [92]. 

Three new compounds, named brunneins A–C (**12**–**14**, Figure 2), were isolated from *Cortinarius brunneus* [26]. Later, brunnein A (**12**) was also found in diverse *Hygrophorus* species [27]. The diastereomer of brunnein B (**15, Figure 2**) was also isolated from *Cyclocybe cylindracea*, and exhibited a marked antioxidant activity [28]. In addition, acid **16** (Figure 2) was isolated from *C. brunneus*, which was the first time from a non-vegetal source [26].

10-Hydroxy-infractopicrin (**17**, Figure 2) was isolated for the first time together with the already-known infractopicrin (**18**, Figure 2) from the toadstool *C. infractus* [29]. Both compounds **17** and **18** inhibited acetylcholinesterase with a higher selectivity than the reference drug galanthamine, thus they were suggested as potential drugs for Alzheimer’s disease.

A new family of 16 compounds, named metatacarbolines, was identified in the fruiting bodies of *Mycena metata* [30]. Each of these compounds is a β-carboline bonded to a specific amino acid, with the exception of metatacarboline A and 6-hydroxymetatacarboline A (**19** and **20**, Figure 2). 6-Hydroxymetatacarboline D (**21**, Figure 2) was the only isolated compound in this study [30], although a later study focused on the synthesis of some metatacarbolines. In particular, the syntheses of metatacarbolines A (**19**) and C–F (**21**–**25**, Figure 2) were reported with 40–75% overall yields [31] and their availability allowed for evaluating their anticancer activity. Metatacarbolines D (**23**) and F (**25**) showed a significant antiproliferative activity by arresting the cell cycle at the sub G0/G1 and G2/M phases of the cell cycle, respectively [31]. 

Flazin (**26**, Figure 2) was isolated from *Suillus granulatus* and *Boletus umbriniporus* for the first time from mushrooms [93]. It is the only reviewed alkaloid containing the β-carboline moiety joined with a furan ring.

The most recent β-carboline discovered in mushrooms (**27**, Figure 2) was isolated from *Sarcomyxa edulis* [32]. Compound **27** is the only reviewed β-carboline with a ketone group located in an exocyclic position. It showed a remarkable anti-inflammatory activity against lipopolysaccharide-induced NO [32].

### 2.2. Pyrroloquinoline Alkaloids

Pyrroloquinolines are a family of natural compounds mostly isolated from marine sponges, which gained interest with the discovery of the cytotoxic alkaloid discorhabdin C (**28**, Figure 3) in 1986 [94,95]. Diverse studies developed during 2002–2022 proved that mushrooms can also be sources of pyrroloquinolines, although a low number of studies on their bioactivities were performed. Many of the pyrroloquinoline alkaloids belong to the family of mycenarubins (**29**–**34**, Figure 3), which were discovered in 2007, with the isolation of mycenarubin A (**29**) [33]. Mycenarubin A was isolated together with its dimer mycenarubin B (**30**) from *Mycena rosea*, which represents the first occurrence of a dimeric pyrroloquinoline alkaloid in nature. The synthesis of mycenarubin A (**29**) was accomplished in 10 steps and produced a 21% total yield by Backenköhler et al. (2018) [39]. Later, mycenarrubins D–F (**31**–**33**) were isolated from *Mycena haematopus* [36]. Mycenarubin A (**29**) was also obtained from *M. haematopus* [34] and *Mycena pelianthina*, with the last species also being a source for the isolation of mycenarubin D (**31**) [35].

Mycenarubin D (**31**) showed an antibacterial activity against *Azovibrio restrictus*, *Azoarcus tolulyticus,* and *Azospirillum brasilense*, whereas mycenarubin A (**29**) was shown to be inactive as an antibacterial compound [34,37]. Thus, the presence of the C=NH unit at position 7 is a key group for the bioactivity of these pyrroloquinoline alkaloids.

Successively, mycenarrubin C (**34**, Figure 3) was isolated from *M. rosea* [37]. Compound **34** is a special pyrroloquinoline alkaloid with an eight-membered ring, which contains an additional C_1_ unit. The same authors also suggested that mycenarubin A (**29**) is the precursor of mycenarubin C (**34**). 

Sanguinones A and B (**35** and **36**, Figure 3) were isolated from *Mycena sanguinolenta*, with sanguinone A (**35**) being the main metabolite [38]. The same article also reported the first isolation of sanguinolentaquinone (**37**, Figure 3), and the identification of decarboxydehydrosanguinone A (**38**, Figure 3) as an oxidative decarboxylation artifact of sanguinone A (**35**). The synthesis of **37** was later realized in eight steps and with a 28% total yield [39].

Haematopodin B (**39**, Figure 3) was isolated from *M. haematopus*, together with the already known haematopodin (**40**, Figure 3) [36]. The authors suggested that haematopodin (**40**) is the degradation product of haematopodin B (**39**). Haematopodin B (**39**) was shown to be as active as the reference antibiotic drug gentamicin against *A. tolulyticus*.

Pelianthinarubins A and B (**41** and **42**, Figure 3), two new pyrroloquinolines isolated from *M. pelianthina*, possess a more complex structure than the usual pyrroloquinoline alkaloids [35]. They might play a role in the chemical defense of *M. pelianthina* [35].

Mycenaflavins A–D (**43**–**46**, Figure 3) were first isolated from the fruiting bodies of *M. haematopus*, with mycenaflavin D (**46**) being the first dimeric pyrroloquinoline alkaloid with a C-C bridge between the two pyrroloquinoline units [34]. Compounds **43**–**45** differ from other pyrroloquinolines by possessing an additional double bond between C-3 and C-4, which generates a yellow color; whereas mycenaflavin D (**46**) is purple due to the extended conjugated π system [34]. The synthesis of mycenaflavin B (**44**) was achieved in eight steps and with a 15% total yield by Backenköhler et al. (2018) [39]. Alkaloid **44** showed a moderate cytotoxicity against fibroblast and melanoma cells [39]. The authors suggested that this bioactivity could be related to the planarity of the compound in relation to the possibility of DNA intercalation [39]. 

### 2.3. Pyrroles 

The structure of pyrroles, with a high electron density in their heteroaromatic ring, is of special interest when developing new bioactive drugs [96]. The alkaloids of this subgroup attract a great interest for their anticancer, antimicrobial, antiviral, antimalarial, antitubercular, anti-inflammatory, and enzyme inhibiting properties [97]. Indeed, according to the Scifinder database, 643 patents that used the term “pyrrole” in biological studies were issued, 550 of them since 2002. Before this date, diverse alkaloids including a pyrrole in their structure were known of mostly from a vegetal or marine origin. In mushrooms, the discovery of sciodole (**47**, Figure 4) from *Tricholoma sciodes* [98], an alkaloid containing both a pyrrole and indole moiety in its structure could be highlighted. 

It is worthy to note that pyrroles show acid properties, so they would not comply with the essential requirement to define them as alkaloids. However, pyrrolizidine alkaloids commonly accumulate as *N*-oxides, which are transformed into pyrrole derivatives during their metabolism [99]. This consideration makes it possible to find in the bibliography pyrrolic compounds cited as alkaloids by their authors, which is reviewed in this section.

From 2002, different pyrroles were discovered from mushrooms. All of them have an aldehyde function at C-2 and a primary hydroxyl or methoxy group at C-5, being structural derivatives of 5-(hydroxymethyl)-1*H*-pyrrole-2-carboxaldehyde (**48**, Figure 4). In fact, **48** was found for the first time from *Inonotus obliquus* in 2014 [100], and successively also from other mushroom species, as will be seen throughout this section.

Inotopyrrole (**49**, Figure 4), a benzyl derivative of **48** isolated from *I. obliquus*, was reported as a new mushroom compound [100]. However, its isolation and structure determination were previously reported when compound **49** was isolated from *Ganoderma capense* and named as ganodine [66,101]. Inotopyrrole B (**50**, Figure 4), a related compound formed by the bonding of the same pyrrole scaffold with an indole, was also found in *I. obliquus* [40]. Both inotopyrrole (**49**) and inotopyrrole B (**50**) were also isolated from the edible mushroom *Phlebopus portentosus* [41]. Structurally, inotopyrrole B (**50**) shares with the aforementioned sciodole (**47**, Figure 4) the particularity of presenting a pyrrole and indole moiety in its structure.

Three carboxylic acids (**51**-**53**, Figure 4) related to this family were isolated from the fruiting bodies of *Leccinum extremiorientale,* with **51** being a new compound. Compounds **51**–**53** showed a poor cytotoxicity [42]. Compound **53** had been already isolated from the plant *Lycium chinense* [44] and successively from the mushroom *Basidiomycetes-X* [43]. Compound **53** showed a remarkable hepatoprotective activity, suggesting that the carboxylic group of this pyrrole plays an important role in this biological activity [44]. 

Phlebopines A–C (**54**–**56**, Figure 4) were discovered in 2018 from *P. portentosus* [41], a species that also produces compounds **49** and **50**. The absolute configuration of phlebopine B (**55**) was not identified. 1-Isopentyl-2-formyl-5-hydroxy-methylpyrrole and 2-[2-formyl-5-(methoxymethyl)-1*H*-pyrrole-1-yl]propanoate (**57** and **58**, Figure 4), which were previously found only from vegetal sources, were also reported as metabolites of *P. portentosus* [41,46]. Phlebopine C (**56**) and compound **58** are closely related, differing only in the length of the alkyl chain of their ester group. Compound **58** showed a relevant inhibitory activity towards pancreatic lipase [46].

The first isolation of pyrrolefronine from *Grifola frondosa* was reported by Chen et al. (2018) [45], although its structure corresponds with that of phlebopine A (**54**). Five other already-known pyrroles (**48**, **49,** and **59**–**61**, Figure 4) and acids **53** and **62** (Figure 4) were also isolated from *G. frondosa* [45]. Pyrrolezanthine (**61**), previously isolated from different vegetal species and later from the fermentation of a fungus with a plant [47], correspond with the phenolic form of inotopyrrole (**49**). An inhibitory activity against α-glucosidase was found for compounds **59**–**62**, specially for compound **61 [45]**, which also showed anti-inflammatory effects [47].

4-[2-Formyl-5-(hydroxymethyl)-1*H*-pyrrol-1-yl] butanamide (**63**, Figure 4), the amide form of **52**, was isolated together with the already-known **48** and the acid **53** from the edible Japanese mushroom *Basidiomycetes-X*. A weak antioxidant activity was described for **63 [43]**. 

### 2.4. Indoles 

Indole alkaloids are of great relevance for drug development. In fact, some natural ones have been approved by the Food and Drug Administration (FDA), such as vincristine, vinblastine, vinorelbine, and vindesine for the treatment of leukemia, lymphoma, melanoma, breast cancer, and non-small cell lung cancer [102]. From a structural point of view, the indole scaffold corresponds to a pyrrole bonded to a benzene. However, unlike pyrrole compounds, for which there are not a remarkably high number of compounds identified in mushrooms, indoles are more abundant in these species. In fact, more than 140 compounds bearing an indole heterocycle were found in mushrooms, with the amino acid L-typtophan being the biogenic source of most of them [103]. Thus, structurally related indoles with endogenous activities such as 5-hydroxy-L-tryptophan, tryptamine, serotonin, melatonin, and bufotenin were identified in a diverse range of mushrooms [103].

Psilocin alkaloid, and its phosphorylated counterpart, psilocybin (**64** and **65**, Figure 5), are among the most studied indole metabolites produced by mushrooms. They are hallucinogens found in mushrooms of the genus *Psilocybe*, *Panaeolus*, *Conocybe*, *Gymnopilus*, *Stropharia*, *Pluteus,* and *Panaeolina* [21], which have been known of since the middle of the last century after their isolation from *Psilocybe mexicana* [104]. Both compounds have been extensively described in recent reviews [21,105,106]. 

Both compounds **64** and **65** have special relevance in therapeutic treatments due to their low toxicity and suitable physiological tolerance [21]. Because of these properties, throughout the period 2002–2022, they have continued to be the object of study in numerous investigations. The results are described in 280 articles and 101 patents that contain the term “psilocin”, as well as 924 articles and 176 patents for “psilocybin“, as can be found for this period in the Scifinder database. Indeed, psilocin-mushrooms containing psilocin (**64**), psilocybin (**65**), and psilacetin (**66**, Figure 5) have been suggested as viable chemotherapeutic agents against SARS-CoV-2 [107]. 

Norpsilocin, baeocystin, norbaeocystin, aeruginascin, and bufotenin (**67**–**71**, Figure 5) are other examples of indoles with psychoactive properties produced by mushrooms, although they have not been studied much other than psilocin or psilocybin [21,108]. The syntheses and biological evaluation of some of these were carried out by Sherwood et al. (2020) [108], and the antiviral activity of bufotenine (**71**) was reported [109]. Norpsilocin (**67**) was isolated for the first time in 2017 (from *Psilocybe cubensis*) and its psychoactive properties, with an agonist activity of the human 5-HT_2A_ receptor close to that of psilocin (**64**), were also described [110]. Recently, effects in time estimation and cognition in in vivo assays for norpsilocin (**67**) were estimated, while both psilocin (**64**) and psilocybin (**65**) produced unspecific effects in these two parameters [111].

In addition to new pharmacological properties, studies of agronomic interest have also been developed with indolic compounds. In fact, the production of 6-hydroxy-1*H*-indole-3-acetamide (**72**, Figure 5), which is an already-known mushroom compound, was recently related to glyphosate resistance [112].

3-Chloroindole (**73**, Figure 6) was isolated from *Hygrophorus paupertinus*, the first time from a terrestrial organism, together with indole (**74**, Figure 6) and was identified as one of the compounds responsible for the fecal odor of this mushroom [113]. However, indole (**74**) is a bicyclic and heterocyclic aromatic compound, and not an alkaloid. 5-Methoxy-4-methoxymethyl-2-methyl-1*H*-indole (**75**, Figure 6) was only found in the volatile components of *Tricholoma caligatum*, by Fons et al. (2006) [48]. Its synthesis was accomplished in two steps and it achieved 17% global yield starting from 5-hydroxy-2-methylindole [114].

The three *N*-glucosylated indoles **76**–**78** (Figure 6) were isolated from the basidiomycete *C. brunneus*, with **76** and **77** being new compounds [49]. The endogenous role of compound **78** was investigated, and it was suggested that it may either act as an inactive transport or storage form of auxin (growth regulator), or that it is a detoxification product [49]. 

Macrolepiotin (**79**, Figure 6) was isolated from *Macrolepiota neomastoidea*, a poisonous mushroom that causes severe gastrointestinal symptoms [50]. 

7-Methoxyindole-3-carboxylic acid methyl ester and 1-methylindole-3-carboxaldehyde (**80** and **81**, Figure 6) were isolated from *Phellinus linteus* [51].

5-Hydroxyhypaphorine (**82**, Figure 6) was isolated for the first time from *Astraeus odoratus*, a species that also produces the betaine hypaphorine (**83**, Figure 6) [52]. 

Echinuline (**84**, Figure 6), an indole alkaloid previously isolated from filamentous fungi and vegetal species, was obtained for the first time from the basidiomycete *Lentinus strigellus* [115]. Alkaloid **84** showed cytotoxicity and damage to the alveolar walls and liver, and food and water containing this compound are refused by animals [116]. It belongs to the family of echinulins, alkaloids whose biosynthesis is currently under study [117,118].

4-(Ethoxymethyl)-1*H*-indole (**85**, Figure 6) was isolated together with its methoxylated derivative **86** (Figure 6) from *Tricholoma flavovirens*. Alkaloid **86** was an already-known indole previously found in other *Tricholoma* species. Both compounds have been shown to be active in plant growth bioassays [53].

Corallocin C (**87**, Figure 6) was isolated for the first time from *Hericium coralloides*. Compound **87** belongs to the family of corallocins, and has been characterized for containing an indole moiety. It showed a remarkable activity for stimulating neurite outgrowth [54].

At this point, it is worth mentioning the previously detailed isolation of inotopyrrole B (**50**, Figure 4) from *I. obliquus*, an alkaloid containing both an indole and a pyrrole moiety in its structure [40].

Terpendoles N and O (**88** and **89**, Figure 6) were isolated as new compounds from *Pleurotus ostreatus* [55]. The last time a new compound of the terpendole family was discovered was in 1999, when terpendole M was isolated from the fungus *Neotyphodium lolii* [119]. It should be noted that during the 2002–2022 period, new studies were carried out evaluating the bioactivity of some terpendoles [120,121,122,123,124,125,126]. Three new terpendoles produced by the fungus *Volutella citronella*, two of them named terpendoles N and O (**90** and **91**, Figure 6), were reported in another study [127], but their structures were different to those published by Zhu et al. (2020) [55]. In fact, the new structure for terpendole N differed significantly because the indole system contains an amide group. Compound **91** induced the inhibition of sterol *O*-acyltransferase isozymes, while **90** was not active [127]. Terpendoles N and O, with respect to most of the alkaloids cited in this review, presented up to eight rings, including two epoxides, and were the only compounds reviewed that contained an epoxide ring in their structure.

### 2.5. Miscellaneous Alkaloids 

The family of dictyoquinazols was discovered in *Dictyophora indusiata* by Lee et al. (2002) [56]. Dictyoquinazols A–C (**92**–**94**, Figure 7) showed a neuroprotective potential against excitotoxicity in cultured mouse cortical neurons. They significantly protected the neurons from glutamate-induced neurotoxicity (at 5–10 µM) and from toxicity induced by *N*-methyl-D-aspartate (at 10–30 µM), although no antioxidant properties were found from the radical scavenging assays. Diverse synthetic strategies obtaining dictyoquinazols were later published [128,129,130]. In particular, the most recent one to synthesize dictyoquinazol A (**92**) [131] also allowed for the preparation of structural analogs of **92** with neuroprotective properties, which were used to carry out a SAR study whose conclusions will be reported in Section 3.

Concavine (**95**, Figure 7), a new rearranged diterpene alkaloid, was isolated from *Clitocybe concava*, and showed a weak antibacterial activity against *Bacillus cereus* and *Bacillus subtilis* [57]. The total synthesis of compound **95** was accomplished in 16 steps and it achieved a 4.2% global yield [132], as well as the synthesis of diverse chlorinated analogs with an improved antibacterial activity [133].

Pyriferines A–C (**96**–**98**, Figure 7), characterized for containing a heterocyclic eight-membered ring, were isolated from the fruiting bodies of *Pseudobaeospora pyrifera* [58]. 

Pycnoporin (**99**, Figure 7), a new phenoxazone alkaloid, was isolated together with the already-known phenoxazones cinnabarin (also named polystictin), tramesanguin, and cinnabarinic acid from *Pycnoporus cinnabarinus*. Compound **99** showed a moderate antitumor activity [59].

The new alkaloid sinensine (**100**, Figure 7) was isolated from the fruiting bodies of *Ganoderma sinense* [60]. This compound was proven to be significantly active as a protecting agent against the injury induced by hydrogen peroxide oxidation on human umbilical cord endothelial cells (protective rate of 70.90% and EC_50_ = 6.2 mmol/L). Successively, sinensines B–E (**101**–**104**, Figure 7) were isolated from the same mushroom, although no studies on the bioactivity of these alkaloids were described [61]. Compounds **103** and **104** only differ in the number of carbon atoms of their oxygenated ring. More recently, sinensine E (**104**) was isolated together with the new alkaloid **105** (Figure 7) from *Ganoderma luteomarginatum*. Both compounds appeared to be a racemic mixture [62]. 

Several new alkaloids (**106**–**124**, Figure 7) were also achieved from the *Ganoderma* species and these findings will be detailed in the following paragraphs. Ganocochlearine A (**106**), the non-hydroxylated form of 105, was isolated together with ganocochlearine B (**107**) from *Ganoderma cochlear* [64]. Ganocochlearine A (**106**) was later isolated from *Ganoderma australe*, showing the protective activity of SH-SY5Y cells from glutamate-induced neural excitotoxicity and, consequently, its potential as a drug against neurodegenerative disorders [65]. Ganocochlearine A (**106**) was also later obtained from *Ganoderma lucidum* and exhibits remarkable neuroprotective (EC_50_ = 2.49 μM) and anti-inflammatory activities (IC_50_ = 4.68 μM) [66].

Two new alkaloids close to sinensine E (**104**), named ganocalicines A and B (**108** and **109**, Figure 7), were isolated from *Ganoderma calidophilum* [67]. Compounds **108** and **109**, which are a methoxylated and non-hydroxylated form of sinensine E (**104**), respectively, were tested in anti-allergic assays. Alkaloid **108** showed its potential as a preventative or relieving drug against allergic symptoms: inhibitory effects on β-hexosaminidase activity (IC_50_ = 9.14 µM) and on the production of the allergic cytokine IL-4 and the lipid mediator LTB4 in antigen-stimulated RBL-2H3 cells (at 5–10 µM) [67].

Ganocochlearines C–I (**110**–**116**, Figure 7) are isolated from *G. cochlear* as racemic or scalemic mixtures [63].

Lucidimines A–D (**117**–**120**, Figure 7), four new alkaloids, were isolated from the fruiting bodies of *G. lucidum* [66,68], with lucidimine C (**119**) also being found in *G. cochlear* [63]. The total syntheses of lucidimines B (**118**) and C (**119**) was realized by Chen and Lan (2018) [69]. The antioxidant properties and relevant antiproliferative activity against MCF-7 cells (EC_50_ = 0.27) of compound **118** were also reported [69]. The poorer or null activities of compound **119** should be attributed to the presence of a methoxy group on the cyclopentene ring which **118** lacks. Lucidimine E (**121**, Figure 7) was successively isolated from the same mushroom and showed a significant anti-inflammatory activity [66].

Ganoapplanatumine A (**122**) and ganoapplanatumine B (**123**), the latter as a racemic mixture, were alkaloids obtained from *Ganoderma applanatum* [70]. Alkaloid **123** was also isolated from *G. cochlear* [63]. 

A new alkaloid, named australine (**124**, Figure 7), a disubstituted pyridine, and two new meroterpenoids, named australins A and B, were isolated together with five known compounds from the fruiting bodies of *G. australe*. The known compounds were identified as lingzhine C; ganocalicine B (**109**); and ganocochlearines A, C, and H (**106**, **110** and **115**). Australine (**124**) and ganocochlearine A (**106**) and showed a significant protection ability against SH-SY5Y cells from glutamate-induced neural excitotoxicity at 10 µM [65]. Previously, a new tetrahydroxy pyrrolizidine alkaloid, named australine, was isolated from the seeds of *Castanospermum australe* and was shown to be a potent and specific inhibitor of amyloglucosidase [134]. However, the two alkaloids have a very different structure. 

Erinacerins M–P (**125**–**128**, Figure 8) were isolated from the medicinal mushroom *Hericium erinaceus* [71]. They showed a moderate cytotoxic activity. Later, erinacerin V (**129**, Figure 8) was described as a new alkaloid purified from the mycelial culture of a unique North American edible *Hericium* mushroom [72].

Rosellin A (**130**) and B (**131**) (Figure 8) were isolated as new glycosylated diketopiperazine alkaloids from the fruiting bodies of *Mycena rosella*, with **130** being obtained in a better yield [73]. Compound **130** showed a herbicidal activity, inducing strong bleaching of the leaves of *Lepidium sativum* [73].

Consoramides A–C (**123**–**134**, Figure 8) were isolated from *Irpex consors* as new zwitterionic alkaloids, together with different stereumamides, including stereumamide D (**138**) [74]. The closely related stereumamides A–D (**135**–**138**, Figure 8), which were the first example of a sesquiterpenes combined with α-amino acids to form quaternary ammonium hybrids, were previously isolated from *Stereum hirsutum*. Stereumamides A (**135**) and D (**138**) showed an antibacterial activity against *Escherichia coli*, *Staphylococcus aureus,* and *Salmonella typhimurium*, with minimum inhibitory concentration (MIC) values of 12.5–25.0 μg/mL [75].

## 3. Structure−Activity Relationship Studies 

Throughout Section 2, more than 100 compounds discovered in 2002–2022 (Table 1) were described, as well as their bioactivity (if it has been evaluated). Section 3 focuses on the most relevant SAR results found assaying the activity of different interrelated compounds, or of the synthetic analogs of the reviewed alkaloids.

Regarding the β-carboline alkaloids, 16 analogs structurally related to harman alkaloids (Figure 2) were synthetized. These analogs presented diverse substituents at positions 1 or 9 (see **139**, Figure 9), and it was found that both type of analogs had an improved broader spectrum of bactericidal activity. An improved activity was observed when the methyl or propyl groups were at C-1, whereas the benzyl group at position 9 could reduce it. On the other hand, all of the analogs showed an insecticidal activity, proving that the modifications applied did not generate a significant improvement in this context [135]. In a later study, a wide group of harman analogs were synthesized to improve the antibacterial activity of this alkaloid. This was achieved by diverse analogs, with **140** being the most active one, which also improved the activity of the positive control. Different SAR conclusions were obtained from this study. The methoxy group at C-6 (see **140**, Figure 9) is beneficial for its antibacterial activity. Furthermore, it was concluded that the type of halogen substituents (CF_3_ > Br > Cl or CH_3_ > F or NO_2_), the position of the halogen atom (*para* > *meta* > *ortho*), and the kind of aromatic substituent R are significant for the antibacterial activity of the tested analogs [136].

Flazin (**26**, Figure 2) is a compound with a weak antiviral activity; thus, a wide collection of analogs to improve this activity were synthesized [137]. The results suggest that certain substituents at positions 3, 1′, and 5′ of flazin (see **141**, Figure 9) might play a key role. The best result was obtained assaying flazinamide (**141**) (therapeutic index of 312.0, and EC_50_ = 0.38 µM). Therefore, the optimal combination is the one provided by the CONH_2_ group at C-3, an *O*-atom in position 1′, and the CH_2_OH group at C-5′. 

SAR results were also obtained after testing the cytotoxicity of β-carboline alkaloids. Thus, the shift of the methoxy group of harmine (**3**, Figure 2) from C-7 to C-4 enhanced the cytotoxic activity; in addition, the substitution of C-1 is essential for achieving high activity levels [138]. Other authors have also reported the potential cholinesterase inhibitory activity of β-carbolines, and it was found that the quaternary ones are about one-sixth as potent as the reference alkaloid physostigmine [26]. A recent review on a wider overview on the bioactivities of β-carbolines and canthinones was recently published by Farouil et al. (2022) [86].

A complete report on SAR studies carried out on pyrrole compounds was published by Ahmad et al. (2018) [139]. As detailed in Section 2.3, this reviewed subgroup of compounds consists of derivatives of 5-(hydroxymethyl)-1*H*-pyrrole-2-carboxaldehyde (**48**, Figure 3). Compound **48** has moderate or low antifungal, antibacterial, and cytotoxic activities, as well as being inactive as an antioxidant or insecticidal compound [140]. It also showed moderate enzyme (α-glucosidase) inhibition [45]. On the other hand, some of the new pyrroles with different substituents on the nitrogen atom (**49**, **54,** and **59**–**62**, Figure 3) significantly improved this activity, especially compound **61**. Thus, it could be pointed out that the higher substitution of this *N* atom favors the inhibition of the tested enzymes. Regarding the anti-proliferative activity against cancer cell lines, these substituted alkaloids did not show improved levels over **48** [45]. These results suggest that it cannot be generalized that the substitution on the *N* atom induces a general activity improvement, which is in accordance with the results observed for other pyrrole compounds reported in the literature [139].

Inotopyrroles (**49** and **50**, Figure 3) possess a remarkable neuroprotective activity, especially **50 [41]**. On the other hand, alkaloids **54**–**58** (Figure 3) showed a lower activity. This result suggests that pyrroles bonded to another aromatic ring may improve this kind of pharmacological activity. It is worth highlighting that, in the case of **50**, this aromatic ring is contained in an indole system. Indeed, pyrrolezanthine (**61**, Figure 3), the phenolic form of **49**, has a strong inhibitory activity (IC_50_ = 28.65 µM) against mammalian α-glucosidase [45], as well as diverse anti-inflammatory effects, sometimes presenting different behaviors according to the concentration [47].

Regarding indole alkaloids, the antifungal activity of the new indoles described in Section 2.4 has not been tested. However, the indole moiety appeared essential for the antifungal activity, as reported for some of the analogs reviewed [141]. Thus, the study of the reviewed mushroom indoles and of new analogs in antifungal bioassays could be of interest. A wide overview of SAR conclusions obtained for indoles, covering many of the most relevant biological activities for the medical field, has been reported by Thanikachalam et al. (2019) [142]. A SAR study regarding the psychoactive activity of psilocybin (**65**, Figure 5) was carried out. In this study, 17 analogs containing different *N*,*N*-dialkyl substituents, and either a 4-hydroxy or 4-acetoxy group, were tested in in vivo bioassays. All of them were highly or moderately active, where bulkier *N*-alkyl groups and *O*-acetylation were found to affect the potency of the 5-HT receptors studied. It was also suggested that the *O*-acetylated compounds may be deacetylated in vivo, which make them act as prodrugs [143]. The SAR results reported by Sard et al. (2005) [144] found that the psilocybin analogs 1-methylpsilocybin (**142**) and 4-fluoro-*N*,*N*-dimethyltryptamine (**143**) (Figure 9) are potential efficient compounds for the treatment of obsessive compulsive disorders. Moreover, 1-methylpsilocin (**144**) would be of interest as it has been described as a selective agonist at the h5-HT_2C_ receptor.

A SAR study was also performed on indoles **90** and **91** (Figure 6) related to the inhibitory activity of the sterol *O*-acyltransferase isozymes of terpendole compounds. Thus, it was concluded that the opening of the A-ring (see **91**) had a negative effect, the presence of a hydroxyl group at the *N*-ring was not relevant, while the isoprenyl residue in the aromatic ring was not essential [127]. 

SAR conclusions were also described for the reviewed miscellaneous alkaloids. A group of analogs of dictyoquinazol A (**92**, Figure 7) were synthesized. Analogs **145**–**150** (Figure 9) equaled or improved the neuroprotective activities of dictyoquinazol A (**92**) against three injury stimuli (L-glutamate, H_2_O_2_, and staurosporine). The results showed that the methoxy groups linked to the benzene rings decreased the glutamate protection, but improved H_2_O_2_ protection; the modification of the heterocycle ring could improve H_2_O_2_ protection without compromising glutamate or staurosporine protection; and changing the hydroxyl group could improve glutamate protection, without compromising H_2_O_2_ or staurosporine protection [131].

Concavine (**95**, Figure 7) is an alkaloid with a weak antibacterial activity [57]. Diverse analogs were synthesized based on the incorporation of a chlorinated aromatic ring in its structure. This modification significantly improved its antibacterial activity [133]. Thus, **151** showed an antibiotic activity against *B. subtilis*, **152** and **153** against *S. aureus*, **154** against *P. fluorescens,* and **152** against *E. coli* (MIC = 6.25 µg/mL for all these cases) (Figure 9) [133]. The most active of the analogs reported was **155**, followed by **156** (Figure 9), both characterized as being acyclic derivatives of concavine with MIC values always between 1.56–12.5 µg/mL for all of the bacterial species tested. 

Cinnabarin (**157**) showed an antitumor activity with an IC_50_ value of 13 µM [59]. It should be noted that other phenoxazones, particularly Phx-1 and Phx-3 (**159** and **160**, Figure 9), are well-studied compounds for the development of anticancer drugs, as reviewed by Zorrilla et al. (2021) [3]. In the SAR study [59], pycnoporin (**99**, Figure 7) showed a moderate antitumor activity, whereas cinnabarinic acid (**158**, Figure 9) was not active. These compounds only differ in one substituent, allowing for concluding that the carboxyl group negatively affected the antitumor activity of this kind of phenoxazone, whereas the presence of the moiety -CH(OCH_3_)OH or -CH_2_OH at C-9 (see **158**, Figure 9) could significantly improve this activity.

## 4. Conclusions

Here, the new alkaloids and related compounds produced by mushrooms since 2002 have been reviewed. Although mushrooms are a source that has not been studied as much as others in this context, it has been found that 114 new compounds with different structures (Table 1) have been isolated and identified. Different studies have shown the promising levels of bioactivity that many of them have, most of which are activities of pharmacological interest, such as antioxidant, anti-inflammatory, neuroprotective, antibacterial, and enzyme inhibition properties. This affords the opportunity to thoroughly explore these new compounds in future studies, in addition to the alkaloids that have been more studied, such as psilocin and its analogs. Furthermore, it is worth highlighting the low amount of references of studies on activities of agronomic interest, for example, aiming at exploring the phytotoxic potential of alkaloids produced by mushrooms. On the other hand, the development of new syntheses that allow for access to alkaloids in sufficient quantities for their study and to the improvement of their biological activity through structural modifications are also of high interest in this field.

For all of these reasons, mushrooms could be viewed as a source of potential active products, thereby potentially leading to further research on them. 

## Figures and Tables

**Figure 1 biomolecules-12-01025-f001:**
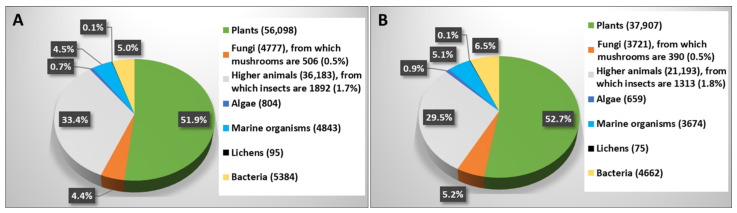
Distribution of references on alkaloids in the database SciFinder (**A**) without a time filter; (**B**) from 2002 to 30 May 2022. References were obtained using the keyword “alkaloid”, plus the corresponding keyword for each type of organism. The number of references for each item are shown in parentheses.

**Figure 2 biomolecules-12-01025-f002:**
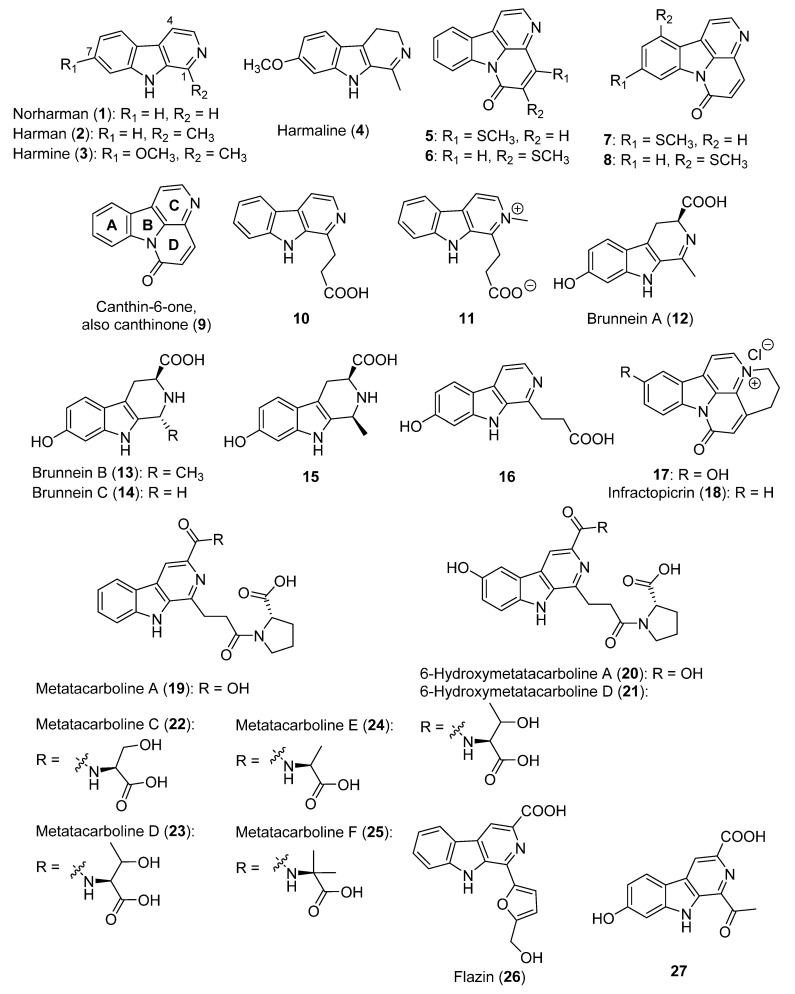
The structures of the β-carboline alkaloids isolated from mushrooms (**1**–**27**).

**Figure 3 biomolecules-12-01025-f003:**
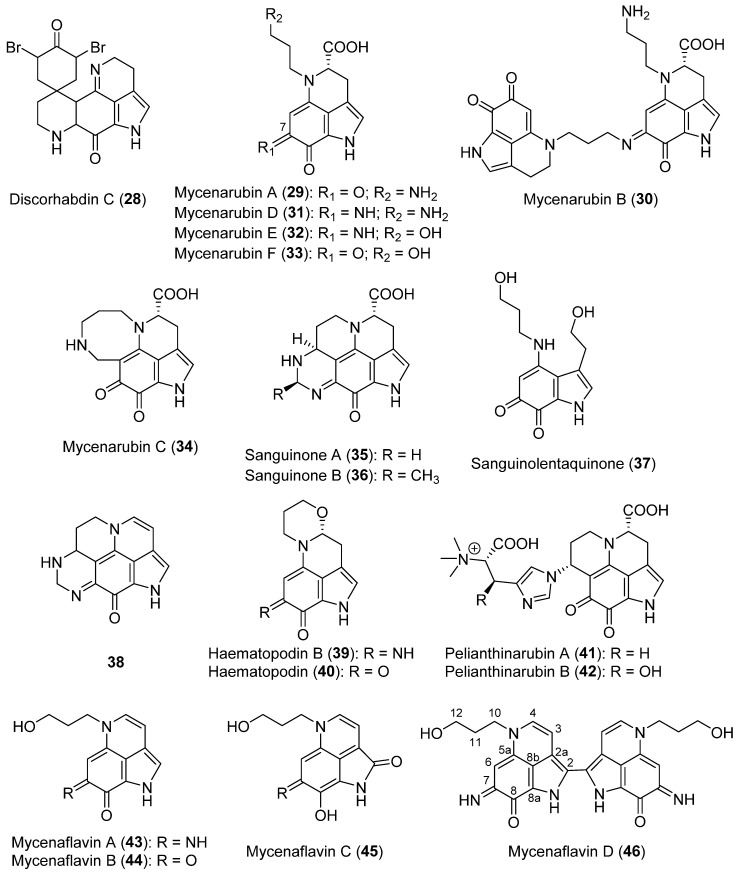
The structures of discorhabdin C (**28**) and the pyrroloquinoline alkaloids isolated from mushrooms (**29**–**46**).

**Figure 4 biomolecules-12-01025-f004:**
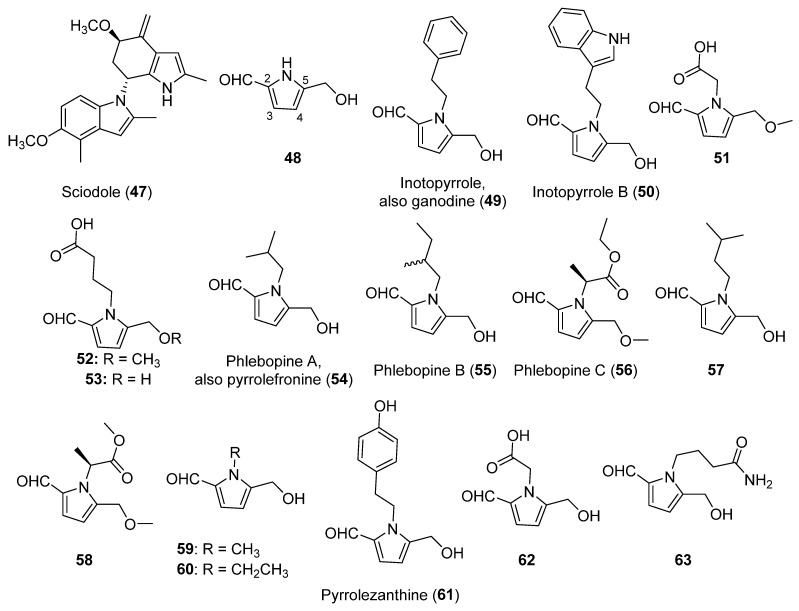
The structures of the pyrrole alkaloids isolated from mushrooms (**47**–**63**).

**Figure 5 biomolecules-12-01025-f005:**
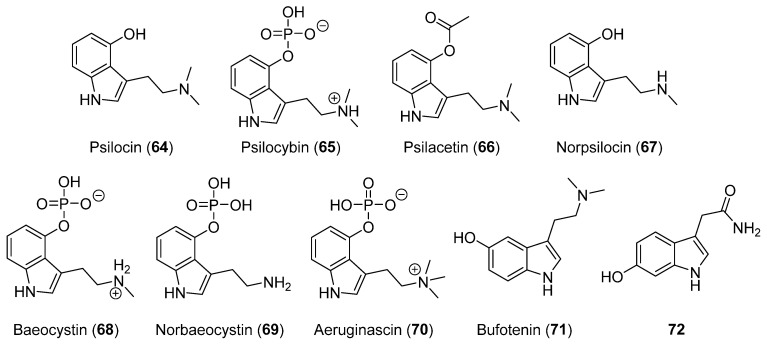
Already-known bioactive indole alkaloids produced by mushrooms (**64**–**72**).

**Figure 6 biomolecules-12-01025-f006:**
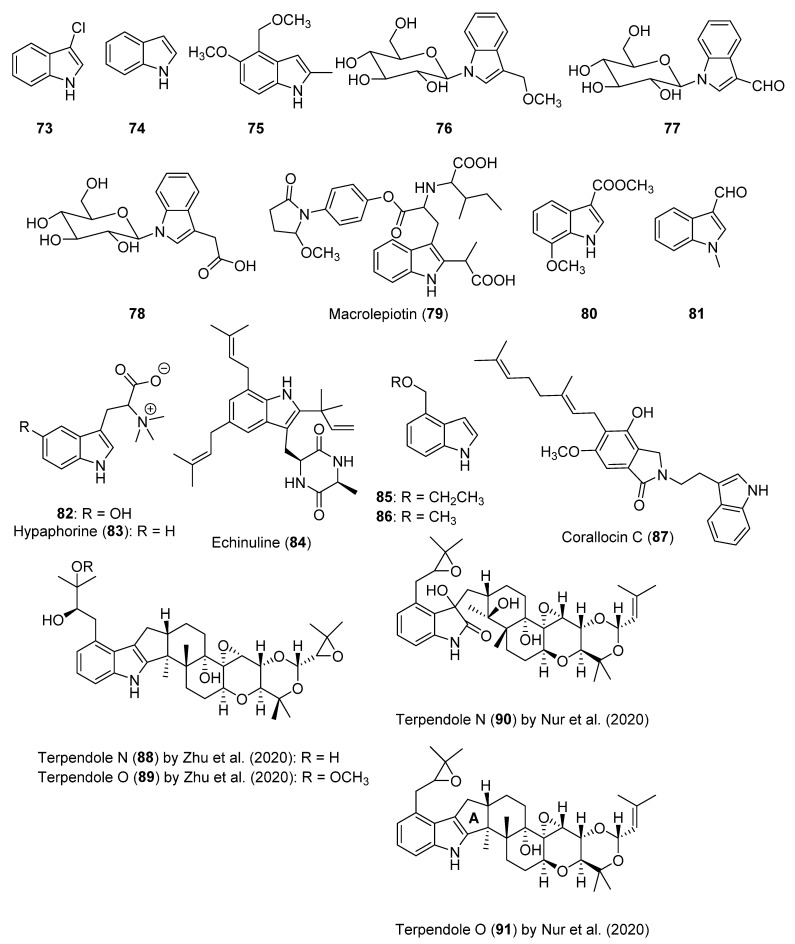
The structures of the indoles and indole alkaloids isolated from mushrooms (**73**–**91**).

**Figure 7 biomolecules-12-01025-f007:**
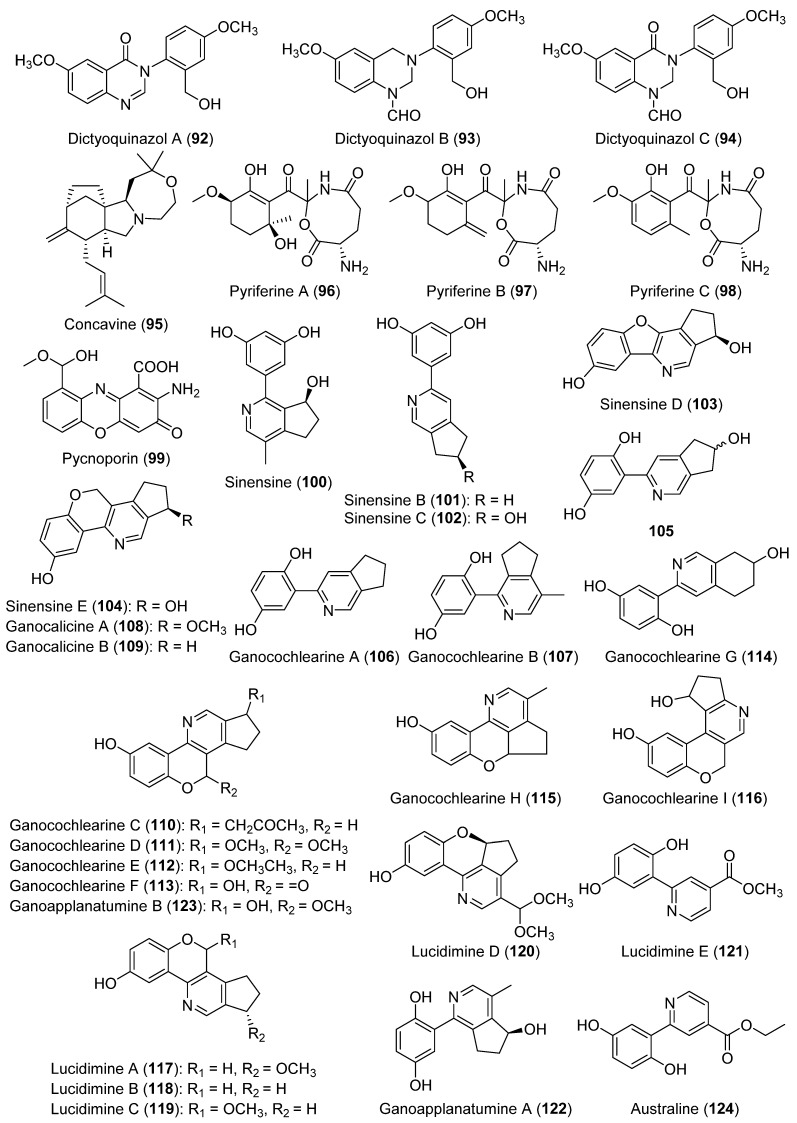
The structures of the miscellaneous alkaloids **92**–**124**.

**Figure 8 biomolecules-12-01025-f008:**
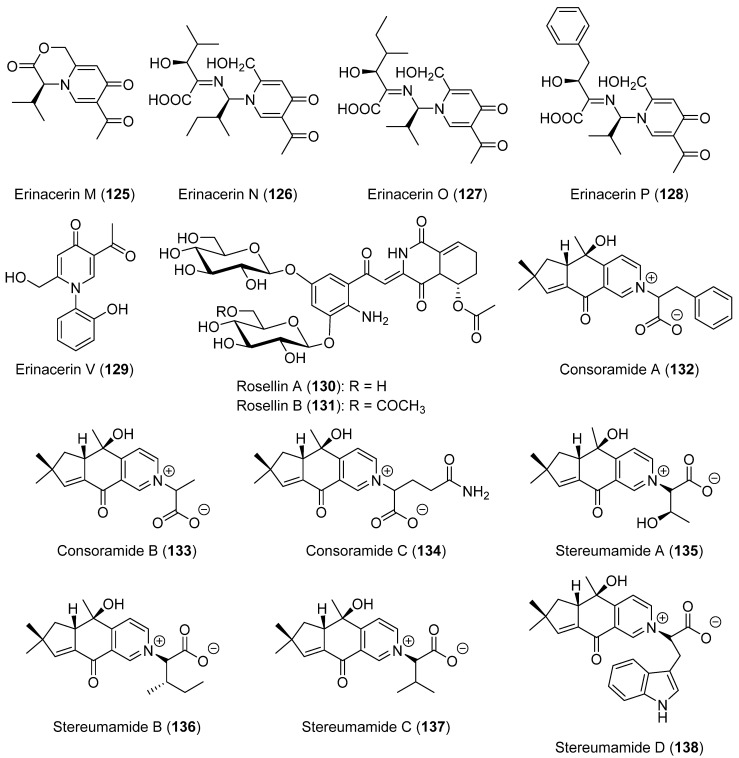
The structures of the miscellaneous alkaloids **125**–**138**.

**Figure 9 biomolecules-12-01025-f009:**
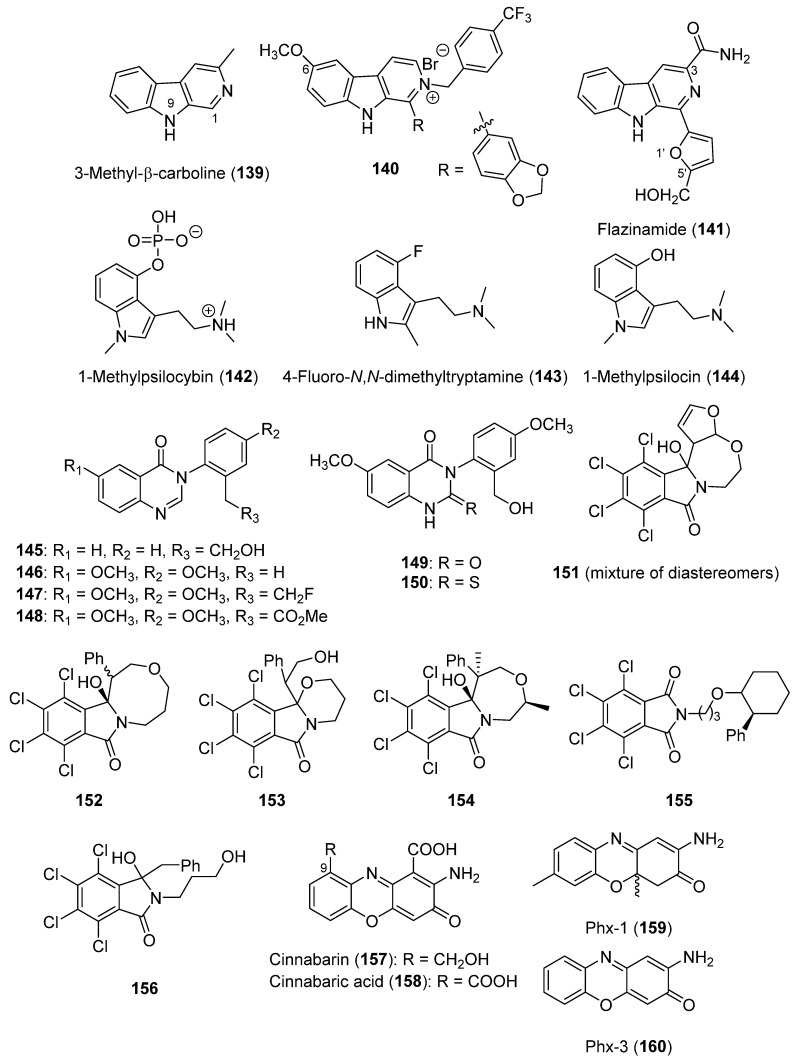
Related compounds and some synthetic structural analogs employed in SAR studies of the reviewed alkaloids **139**–**160**.

**Table 1 biomolecules-12-01025-t001:** Alkaloids and related compounds produced by mushrooms discovered in the period covered by the review (2002–2022).

Alkaloid	Mushroom Source	Biological Activity	References
Subgroup: β-Carboline alkaloids (Figure 2)			
4-(Methylthio)canthin-6-one (**5**); 5-(methylthio)canthin-6-one (**6**); 9-(methylthio)canthin-6-one (**7**); 11-(methylthio)canthin-6-one (**8**); 2-methyl-β-carbolinium-1-propanoate (**11**)	*Boletus curtisii*	-	[25]
Brunnein A (**12**)	*Cortinarius brunneus* Different *Hygrophorus* spp.	-	[26,27]
Brunnein B (**13**); brunnein C (**14**)	*C. brunneus*	-	[26]
C-1 diastereomer of brunnein B (**15**)	*Cyclocybe cylindracea*	Antioxidant	[28]
10-Hydroxy-infractopicrin (**17**)	*Cortinarius infractus*	Inhibition of acetylcholinesterase	[29]
Metatacarboline family (**19**–**25**)	*Mycena metata*	Anticancer, for metatacarbolines D (**23**) and F (**25**)	[30,31]
1-Acetyl-7-hydroxy-9*H*-pyrido [3,4-*b*]indole-3-carboxylic acid (**27**)	*Sarcomyxa edulis*	Anti-inflammatory	[32]
Subgroup: Pyrroloquinoline alkaloids (Figure 3)			
Mycenarubin A (**29**)	*Mycena haematopus*, *Mycena pelianthina* and *Mycena rosea*	-	[33,34,35]
Mycenarubin B (**30**)	*M. rosea*	-	[33]
Mycenarubin D (**31**)	*M. haematopus*	Antibacterial	[36]
Mycenarubin E (**32**); mycenarubin F (**33**)	*M. haematopus*	-	[36]
Mycenarubin C (**34**)	*M. rosea*	-	[37]
Sanguinone A (**35**); sanguinone B (**36**); sanguinolentaquinone (**37**); decarboxydehydrosanguinone A (**38**)	*Mycena sanguinolenta*	-	[38]
Haematopodin B (**39**)	*M. haematopus*	Antibacterial	[34,36]
Pelianthinarubin A (**41**); pelianthinarubin B (**42**)	*M. pelianthina*	-	[35]
Mycenaflavin A (**43**)	*M. haematopus*	Moderate antibacterial	[34]
Mycenaflavin B (**44**)	*M. haematopus*	Moderate antibacterial and cytotoxic	[34,39]
Mycenaflavin C (**45**); mycenaflavin D (**46**)	*M. haematopus*	-	[34]
Subgroup: Pyrrole alkaloids (Figure 4)			
Inotopyrrole B (**50**)	*Inonotus obliquus* and *Phlebopus portentosus*	Neuroprotective against H_2_O_2_ damage	[40,41]
2-[2-Formyl-5-(methoxymethyl)-1*H*-pyrrol-1-yl]acetic acid (**51**)	*Leccinum extremiorientale*	Low cytotoxic	[42]
4-[2-Formyl-5-(hydroxymethyl)-1*H*-pyrrol-1-yl] butanoic acid (**53**)	*Basidiomycetes-X*, *Grifola frondosa* and *L. extremiorientale*	Hepatoprotective, low inhibition of α-glucosidase and low cytotoxic	[42,43,44,45]
Phlebopine A, also pyrrolefronine (**54**)	*G. frondosa* and *P. portentosus*	Inhibition of α-glucosidase, and mild neuroprotective against H_2_O_2_ damage	[41,45]
Phlebopine B (**55**); phlebopine C (**56**); 1-isopentyl-2-formyl-5-hydroxy-methylpyrrole (**57**)	*P. portentosus*	Moderate or mild neuroprotective against H_2_O_2_ damage	[41]
2-[2-Formyl-5-(methoxymethyl)-1*H*-pyrrole-1-yl]propanoate (**58**)	*P. portentosus*	Inhibition of pancreatic lipase activity, and mild neuroprotective against H_2_O_2_ damage	[41,46]
5-Hydroxymethyl-1-methyl-1*H*-pyrrole-2-carbaldehyde (**59**); 5-hydroxymethyl-1-ethyl-1*H*-pyrrole-2-carbaldehyde (**60**); 5-hydroxymethyl-1-acetic acid-1*H*-pyrrole-2-carbaldehyde (**62**)	*G. frondosa*	Inhibition of α-glucosidase	[45]
Pyrrolezanthine (**61**)	*G. frondosa*	Anti-inflammatory and strong inhibition of α-glucosidase	[45,47]
4-[2-formyl-5-(hydroxymethyl)-1*H*-pyrrol-1-yl] butanamide (**63**)	*Basidiomycetes-X*	Weak antioxidant	[43]
Subgroup: Indole alkaloids (Figure 6)			
5-Methoxy-4-methoxymethyl-2-methyl-1*H*-indole (**75**)	*Tricholoma caligatum*	-	[48]
1-(1-β-Glucopyranosyl)-3-(methoxymethyl)-1*H*-indole (**76**); 1-(1-β-glucopyranosyl)-1*H*-indole-3-carbaldehyde (**77**)	*C. brunneus*	-	[49]
Macrolepiotin (**79**)	*Macrolepiota neomastoidea*	-	[50]
7-Methoxyindole-3-carboxylic acid methyl ester (**80**); 1-methylindole-3-carboxaldehyde (**81**)	*Phellinus linteus*	-	[51]
5-Hydroxyhypaphorine (**82**)	*Astraeus odoratus*	-	[52]
4-(Ethoxymethyl)-1*H*-indole (**85**)	*Tricholoma flavovirens*	Plant growth	[53]
Corallocin C (**87**)	*Hericium coralloides*	Stimulation of neurite outgrowth	[54]
Terpendole N (**88**); terpendole O (**89**)	*Pleurotus ostreatus*	-	[55]
Subgroup: Miscellaneous alkaloids (Figures 7 and 8)			
Dictyoquinazols A–C (**92**–**94**)	*Dictyophora indusiata*	Neuroprotective	[56]
Concavine (**95**)	*Clitocybe concava*	Weak antibacterial	[57]
Pyriferines A–C (**96**–**98**)	*Pseudobaeospora pyrifera*	-	[58]
Pycnoporin (**99**)	*Pycnoporus cinnabarinus*	Moderate antitumoral	[59]
Sinensine (**100**)	*Ganoderma sinense*	Protective against H_2_O_2_ oxidation	[60]
Sinensines B-D (**101**–**103**)	*G. sinense*	-	[61]
Sinensine E (**104**)	*Ganoderma cochlear*, *Ganoderma luteomarginatum,* and *G. sinense*	-	[61,62,63]
(+)-6*S*-Hydroxyganocochlearine A and (−)-6*R*-hydroxyganocochlearine A (**105**)	*G. luteomarginatum*	-	[62]
Ganocochlearine A (**106**)	*Ganoderma australe*, *G. cochlear,* and *Ganoderma lucidum*	Neuroprotective and anti-inflammatory	[64,65,66]
Ganocochlearine B (**107**)	*G. cochlear*	-	[64]
Ganocalicine A (**108**)	*Ganoderma calidophilum*	Anti-allergic	[67]
Ganocalicine B (**109**)	*G. australe* and *G. calidophilum*	-	[65,67]
Ganocochlearine C (**110**); ganocochlearine H (**115**)	*G. australe* and *G. cochlear*	-	[63,65]
Ganocochlearines D-F (**111**–**113**); ganocochlearine I (**116**)	*G. cochlear*	-	[63]
Lucidimine A (**117**); lucidimine D (**120**)	*G. lucidum*	-	[66,68]
Lucidimine B (**118**)	*G. lucidum*	Antioxidant and antiproliferative	[66,68,69]
Lucidimine C (**119**)	*G. cochlear* and *G. lucidum*	Antioxidant	[63,66,68,69]
Lucidimine E (**121**)	*G. lucidum*	Anti-inflammatory	[66]
Ganoapplanatumine A (**122**)	*Ganoderma applanatum*	-	[70]
Ganoapplanatumine B (**123**)	*G. applanatum* and *G. cochlear*	-	[63,70]
Australine (**124**)	*G. australe*	Neuroprotective	[65]
Erinacerins M–P (**125**–**128**)	*Hericium erinaceus*	Moderate cytotoxic	[71]
Erinacerin V (**129**)	*Hericium* sp.	-	[72]
Rosallin A (**130**)	*Mycena rosella*	Herbicidal	[73]
Rosallin B (**131**)	*M. rosella*	-	[73]
Consoramides A–C (**132**–**134**)	*Irpex consors*	-	[74]
Stereumamide A (**135**)	*Stereum hirsutum*	Antibacterial	[75]
Stereumamide B (**136**); stereumamide C (**137**)	*S. hirsutum*	-	[75]
Stereumamide D (**138**)	*I. consors* and *S. hirsutum*	Antibacterial	[74,75]

## Data Availability

Not applicable.

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
