# Peer review of "Structures and Biological Activities of Alkaloids Produced by Mushrooms, a Fungal Subgroup"

_biomolecules, 2022, doi:10.3390/biom12081025_

Round 1

Reviewer 1 Report

I reviewed the manuscript "Alkaloids produced by mushrooms: isolation, structures and bioactivities" submitted to the journal Biomolecules. The topic of the manuscript is interesting, but I have several concerns that the authors should answer and modify. As it stands, the manuscript is difficult to read and understand.
L 30-35, I see no need to mention these plant alkaloids in any text. Delete this. Authors only mention some random alkaloids from plants.
In Figure 1, the authors use an unusual (and to me, nonsensical) method of subdividing living organisms. For example, fungi and mushrooms separated (mushrooms are fungi), then animals and insects separated (insects are animals), and algae and marine organisms separated. And then lichens, which are symbiotic organisms made of a fungus and an alga.
So it's not entirely clear to me what mushrooms means to the authors.
I am not quite sure if the authors did the literature search correctly. In an article dealing with food, the words mushrooms are usually used, but in biological research, the authors prefer the term macrofungi (or simply fungi also for mushrooms). So if the authors only used the word fungi for the search, I think they missed a whole lot of literature.
The authors should add a table with the data on the total amount of alkaloids in mushrooms in different species. There are quite a lots of publications of this type. And they should indicate whether these numbers are higher or lower than, for example, in plants and how the amout differ within the species.
In a section 2. New alkaloids in mushrooms since 2002, the authors list only the alkaloids, give their structure and mention some biological activities. It would be better to make a table with the structure, species from which they are isolated and biological activity, and then the authors can compare different structures, etc. in the text. The rest of the text in this section is also difficult to understand. It is a review article and the authors should make the tables/figures with the summarized data to make it easier to read. The way the text is presented is difficult to understand and follow. Also the species from which the alcaloids are isolated should be indicated and discussed biological background of specufic alcaloids distribution.
In section 3, the authors should use the whole word for SAR - structure-activity-relationship in a subtitle.
In the conclusions, a figure should be included that summarizes all the data presented in a review. E.g., with the number of compounds, their biological activity, and SAR.

Reviewer 2 Report

1) It is not clear how “mushrooms” were differentiated from “funi”. In fact, mushrooms are just relatively large fruit bodies of a number of fungal species. Thus, the authors should at least define if the review is only about alkaloids found in fruit bodies of fungi or whether it also includes alkaloids produced by fungal species forming huge fruit bodies irrespective if they are produced in the fruit body or in the mycelium. I addition, there is also a huge number of fungi forming tiny fruit bodies. How to differentiate the topic of the review from them. It would be good if the authors could provide a sound explanation for the criteria for inclusion and exclusion. I hope the criterium was not presence of the word “mushroom” or something like that in a database search because this is for sure not a scientific criterion (see line 67...).

Moreover, I find the title misleading: the authors did not talk about isolation strategies and methods, thus the word “isolation” should be deleted from the title.

2) What was the rationale for limiting the review to the last 20 years? It would be good to explain that, particularly since a comprehensive review describing all compounds would only be slightly lengthier:  the authors stated that 390 of 506 references were published in the last 20 year (line 59).

3) Between section 1 and 2 I am missing a paragraph describing groups of alkaloids present in “mushrooms”

3) Line 33: the sentence “Among the plant alkaloids it is not possible to report those produced by hundreds of species of Amaryllidaceae, which are divided into 12 subgroups based on their carbon skeleton” sounds strange. Why is it not possible to report those alkaloids? The sentence should be revised.

4) I do not agree that a chronological order makes sense. I think re-structuring section 2 after structural or biosynthetic considerations would make more sense. Alternatively, section 2 could be structured after phylogenetic considerations of the species producing the alkaloids.

5) I am not convinced that reporting negative results that excessive as in section 2 is useful. Moreover, a statement “no antibacterial activity was observed” is quite useless since only a limited number of bacteria were tested. I think it would be better to focus on confirmed activities.

6) The legends to Figure 2 and Figure 3 are not informative.

7) The heading of section 2.1 must be corrected.

8) Line 174: “Both compounds 39 and 40 are the only new glycosylated alkaloids produced by mushroom found for this review.” However, also compounds 123 to 125 are glycosylated.

8) Line 193-194: The word “activity” is missing.

9) Line 196: “could represent” Why could? Are they the best studied compounds or are other compounds studied in more detail?

10) Line 198: “However, these two compounds may be the causal agent of some deseases, including Parkinson and cancer” are for sure not the main causal agents for the mentioned disease and thus the sentence should be re-written, e.g. “However, these two compounds might cause  Parkinson and cancer”.

11) Line 297: is the possible DNA intercalation activity purely speculative or is it based on experimental data? Please describe in more detail.

12) Figure 5: R is missing in structures 86/87

13) Line 398: “All of them have been pyrroles with an aldehyde function at C-2,…” might be written: “All of them have an aldehyde function at C-2,…”

14) Line 339: “Compounds 100 showed…” should be written “Compound 100 showed…”

15) Line 356: the “H” In 4-[2-formyl-5-(hydroxymethyl)-1H-pyrrol-1-yl] should be written italic.

16) Line 405: in my point of view indole is not an alkaloid but a normal metabolite.

17) Line 449: “…reported in other study…” should read “…reported in another study...”

Figure 9: names of the compounds or their lead compounds might be included.

18) Section 3: is there any reason for presenting the compounds in this order?

19) Line 458: “Section 3 is focused on…” might be written “Section 3 focuses on…”

20) Line 463: The sentence is difficult to read and grammatically not correct; please revise.

21) Line 540 “Any antifungal activity has been described for the new indoles described in section 2.4.” sounds strange; please revise.

Round 2

Reviewer 1 Report

The authors answer questions to some degree, however, we disagree with some of the answers. As a biologist and someone who has experience with bioactive compounds in fungi, I disagree with the classification in Figure 1, but the authors do not want to accept suggestions. As I said, the classification in Figure 1 makes no sense to biologists. Fungi are kongdom of organisms which include yeasts, rusts, molds, mildews, and mushrooms. So, it is not clear to me what mushrooms mean and what fungi mean. Does fungi include the data for mushrooms? Also higher animals and insects are also still separated. Please, consult biologiests..

Also, the authors should delete the statement I mentioned in " L 30-35. I see no need to mention these plant alkaloids in any text. Delete this. The authors only mention some random alkaloids from plants."

All in all, the authors just add a table and didi not accept the other suggestions and I will let the editor decide what to do with the manuscript. 

Reviewer 2 Report

The manuscript was greatly improved. The order is now more logic and Table 1 is helpful.

However, I think the authors should include an explanation about “differentiation” of “fungi” and “mushrooms” in the introduction, similar to the explanation they provided in the response to the comments of the reviewers. Apart from this the manuscript is ok.
